# Antimicrobial Peptides as Probes in Biosensors Detecting Whole Bacteria: A Review

**DOI:** 10.3390/molecules25081998

**Published:** 2020-04-24

**Authors:** Éric Pardoux, Didier Boturyn, Yoann Roupioz

**Affiliations:** 1Univ. Grenoble Alpes, CNRS, CEA, IRIG, SyMMES, 38000 Grenoble, France; eric.pardoux@protonmail.com; 2Univ. Grenoble Alpes, CNRS, DCM, 38000 Grenoble, France; didier.boturyn@univ-grenoble-alpes.fr

**Keywords:** antimicrobial peptides, bacteria detection, biosensors, surface chemistry, diagnostics

## Abstract

Bacterial resistance is becoming a global issue due to its rapid growth. Potential new drugs as antimicrobial peptides (AMPs) are considered for several decades as promising candidates to circumvent this threat. Nonetheless, AMPs have also been used more recently in other settings such as molecular probes grafted on biosensors able to detect whole bacteria. Rapid, reliable and cost-efficient diagnostic tools for bacterial infection could prevent the spread of the pathogen from the earliest stages. Biosensors based on AMPs would enable easy monitoring of potentially infected samples, thanks to their powerful versatility and integrability in pre-existent settings. AMPs, which show a broad spectrum of interactions with bacterial membranes, can be tailored in order to design ubiquitous biosensors easily adaptable to clinical settings. This review aims to focus on the state of the art of AMPs used as the recognition elements of whole bacteria in label-free biosensors with a particular focus on the characteristics obtained in terms of threshold, volume of sample analysable and medium, in order to assess their workability in real-world applications.

## 1. Introduction

Resistance of bacteria to conventional antibiotics has been growing over the past few years, now making it a global concern, acknowledged by the World Health Organization [1]. Bloodstream infections caused by bacteria in blood have been reported to cause about 5.3 million deaths annually in high-income countries [2]. Among the various solutions to this crisis are antimicrobial stewardship, better diagnostics and, of course, the search for novel and alternative antibiotics.

The discovery of new drugs is currently slowing down [3]. Antimicrobial peptides (AMPs) have been highlighted as potent candidates suitable for new therapeutic strategies against pathogenic micro-organisms [4]. AMPs are a component of the immune system of many organisms, ranging from bacteria [5] to mammals [6], as well as plants [7] and insects [8]. They are defined by their ability to interact with bacteria whether by inhibiting their growth or by lysing them. They can thus be seen as one of the first barriers of innate immunity. AMPs are predominantly short and cationic, however, a wide variety of sequences and structures exist [9]. Sequences and structures can be retrieved from several databases [10]. Despite this variety, when it comes to interactions with bacterial cells, AMPs rely on different action modes. They can be classified in three main categories: (1) membrane-targeting AMPs, (2) AMPs targeting intracellular bacterial components and (3) AMPs inducing a resistance in bacteria [11]. Development of new drugs consists in favouring the killing modes whilst avoiding the induction of resistances. These mechanisms vary from one AMP to another and can also change depending on the bacterial strain which is targeted [12]. However there is a common ground among bacteria killing mechanisms: an initial interaction between the peptide and the bacterial membrane has to happen before the growth inhibition or lysis of the bacterium [13]. This initial interaction is generally intermediated by electrostatic forces between the cationic AMPs and the negatively charged bacterial membrane. Nonetheless, many anionic antimicrobial peptides have been described [14]. Indeed, the presence of hydrophobic amino-acid residues may also help in the insertion of the peptide inside the membrane. An interplay between the electrostatic and hydrophobic interactions is needed for AMPs to interact with bacterial membranes [15]. Molecular targets in the bacterial outer membrane are often considered to be (lipo)teichoic acids in the case of Gram-positive bacteria and lipopolysaccharides (LPS) for Gram-negative ones [16]. Moreover, once the AMPs are at closer range, hydrogen bonding can enhance the adhesion between the peptide and the bacterial membrane [17]. Once this primary interaction achieved, several mechanisms can take place. Synergistic interplay between peptides can occur and either provokes the lysis of the bacterium or the insertion of the AMPs inside the bacterial cell. Not only AMPs have multiple modes of action, but they are also easy to chemically synthesize and only present a low risk of developing resistances in bacteria [18,19]. These characteristics are advantages when considering AMPs as potential new drugs for fighting conventional antibiotic resistance [20].

Nevertheless, as previously noted, efficient drugs and their good use through antimicrobial stewardship programs are not the only key factor to decrease the burden of bacterial resistance. There is also a need for quick and reliable methods of detection of the pathogens causing infections [21,22,23]. Rapid adaptation of the antibiotherapy diminishes both the mortality rates due to bacterial infections and the risk of bacterial resistance to inadequate drugs.

Classically such diagnostics are performed through culture-based methods of samples like blood, saliva or urine. Despite being the gold standard, they present some disadvantages. They are labour-intensive and time consuming, sometimes requiring several days before completing the enrichment of the sample in order to identify the pathogen. Identification techniques—which can rely on microscopy, biochemical assays or immunological analyses—often need trained staff to be performed. This handling may be dangerous since the enriched analyte samples are in that case loaded with pathogenic bacteria at high concentrations.

Beyond these traditional techniques, a multitude of attempts have been developed to identify the pathogens. Two main approaches are based on polymerase chain reaction (PCR) and mass spectrometry (MS). The first one is a method based on genetic analysis of the sample in order to identify the presence of some specific sequences [24,25]. These peculiar strands of nucleic acids are targeted and amplified in order to be detected. Although this technique is able to precisely identify a given subset of pathogens, it still requires costly equipment and tailored probes to detect all possible pathogens. On the other hand, MS-based diagnosis rely on the profiling of the molecular composition of lysates issued from the sample [26]. Identification of the pathogen is given by comparing the molecular data obtained in mass spectrometry to a library of already known profiles. However, as effective as these techniques may be, they remain expensive and require a lot of handling, usually added to complex processes of sample extraction and purification. Researchers are therefore looking for more adapted technical solutions, with the aim to develop fast, affordable, simple to implement and reliable diagnostics.

In this context, biosensors are an exciting alternative to detect whole viable bacterial cells. These devices can be defined as integrated systems containing a biochemical sensing element, able to recognize a biological target through a transduction mechanism. The multiplexing probes that can be integrated, allow a rich variety of targets to be detected, along with high ease of operation and low-cost production. Biosensors can thus be complementary to already existing techniques for the characterization of bacteria. Furthermore, the detection of whole bacteria ensures lower risks of false-positive results that could be provoked by residual components of bacteria inside samples. Yet, in order to obtain an universal sensor able to detect any species of bacteria, it is needed to have adequate and specific receptors [27]. Two main alternatives can be mentioned: either developing arrays of probes, each one being specific to a set or subset of pathogens; or profiling fingerprints of data obtained from pathogens on biosensors functionalized with broad spectrum probes.

AMPs present both characteristics. Some sequences have been described as specific to certain bacterial species or characteristic features (e.g., Gram staining), while other AMPs can interact with any bacteria, although with varying degrees. This diversity can be employed both to design highly specific ligands or wide-spectrum probes able to interact with many different species. A brief survey of literature (Table 1) shows some of the AMPs incorporated in biosensors. A wide spectrum of specificities and affinities is demonstrated. A same AMP—for instance magainin I—can change in terms of specificity depending on the conditions in which it is incorporated. When aiming for wide-spectrum detection of bacteria, combination of several AMPs on the same chip could enable not only universal recognition of pathogens but also their identification. Multiplexing is thus required to achieve such biosensors.

The possibility to design label-free sensors based on AMPs is interesting in order to decrease the complexity of detection protocols. Label-free sensing requires no tags nor intermediary molecules to perform the pathogen recognition. As a result, label-free devices can demonstrate high integrability into other systems. We have therefore arbitrarily chosen to limit the reported studies to label-free systems.

This review thus intends to present the advances made in the field of label-free biosensors based on antimicrobial peptides as recognition molecules for whole bacteria. An introduction to the influence of surface functionalization on the activity of AMPs will be given, before focusing on the state of the art of AMP-based bacterial biosensors. Upcoming challenges and perspectives of research for such sensors will subsequently be discussed.

## 2. Antimicrobial Peptides as a Mean to Detect Bacteria

Even though AMPs are most often used to take benefit of their bactericidal activity, their capacity to preferentially interact with bacterial membranes has opened up the possibility to detect pathogens. Historically, this use was initiated in the field of medical imaging, with the application of fragments of ubiquitin coupled with ^99m^Tc as an isotopic tracer [36]. The AMP fragments target bacterial walls, which allows to distinguish a bacterial infection from an inflammation [37]. Selecting adequate peptidic fragments hence allows to improve the tracing of infections directly in tissues while limiting the bactericidal activity of the AMP. Such application can be useful for both diagnostics and surgery. Recent literature review from Welling and colleagues provides a more comprehensive insight into the current state of research in this area [38]. Notwithstanding these applications of antimicrobial peptides as tracers, this review focuses on the biosensors that incorporate AMPs as the recognition elements in order to detect and/or identify bacterial pathogens in liquid media. Hybrid techniques that would for instance combine AMPs with other biomolecules (e.g., antibodies, aptamers…) are therefore excluded from this review, however some of them will still be discussed as they represent key milestones in this domain.

### 2.1. Inhibiting Bactericidal Activity of AMPs as Probes

When using AMPs as molecular probes, limiting or overpowering the bactericidal activity while keeping the affinity for bacterial membranes is primordial. It is necessary to avoid adverse impacts such as lysis or killing in order to obtain a sensitive detection. This can be achieved mainly through two ways: either by using the recognition domain of the AMP sequence [39] or by modulating the way the peptide is tethered in order to hinder the killing abilities of the AMP. Separately or simultaneously, both approaches can be applied in biosensors (Figure 1).

### 2.2. Determination of Membrane Binding Fragments

Using the recognition fragments of natural AMPs obviously requires one to locate the domains or residues that are essential to the interaction with the bacterial membrane. This can be done through several methodologies. First example can be alanine scanning (or Ala-scan) [40,41], which sequentially substitutes residues one by one with alanine residues to determine the most influential ones. Synthesizing unmodified fragments thanks to the SPOT technique is another way to massively screen peptide variants [42]. This technique relies on the parallel synthesis of several different short peptides arrayed on flexible membranes such as cellulose [43]. The resulting peptides can then be used to study the structure-activity relationship of the AMP. Assaying numerous alternative versions of a single AMP has historically been the main way to decipher their mechanisms and therefore improve them for drug design and other uses [44]. However, it remains tedious to study peptides that way, since it implies several labour-intensive chemical syntheses and subsequent tests. Biosensors often use surfaces on which probes are anchored. This particularity can be beneficial to design functional surfaces able to trap bacteria without killing them.

### 2.3. Exploiting Opportunities Given by Surface Tethering

When designing AMP-based biosensors, immobilization of the peptides onto a surface can be a way to limit their bactericidal activity. The lytic properties of the peptide can thus be inhibited whilst the binding abilities are conserved. This is achievable without necessarily knowing what are the corresponding recognition fragments in the AMP sequence. Although this can lead in some cases to antibacterial surfaces [45,46], it is possible to modulate the killing activity of anchored molecules through varying the linker to the surface. Furthermore, in solution many AMPs require a threshold density of membrane-bound molecules before triggering the disruption on the bacterial membrane [47]. Tethered AMPs could no longer reach that disruption threshold. This latter is often linked to the self-organization of AMPs on the bacterial membrane, in order to create pores for instance. Tethering AMPs hinders such self-organization processes. Nonetheless, it has been shown that tethering can reduce the antibacterial activity without losing membrane interaction properties [48,49]. Modulations of the immobilization technique can produce different properties depending on the nature and length of the linker to the surface. Bactericidal, bacteriostatic or bacteria trapping surfaces can thus be designed. Trade-offs in terms of flexibility and proximity to the surface have to be made depending on the function desired for the surface [50]. The density of grafted AMPs also plays a prominent role in tuning the properties of the surface [51]. Orientation of the peptide once tethered is another important factor in the properties that will be obtained for the surface [52,53,54,55]. No clear consensus exists so far concerning the best way to obtain the most efficient trapping surface for bacteria [50]. Each AMP sequence will have peculiar properties in terms of bactericidal activity as well as in terms of affinity for bacterial membranes. Lastly, it is important to emphasize that AMPs bound to bacterial membranes are not necessarily able to disrupt them. Assaying the resulting device is hitherto the best way to identify the proper way to limit the bactericidal potency of AMPs while obtaining efficient capture properties. The variety of strategies that can be employed in order to modulate the properties of tethered AMPs have been addressed in several reviews [56,57].

## 3. On the Use of AMPs as Ligands for Biosensors

### 3.1. First Applications of AMPs in Biosensors

As far as we know, the first studies on the incorporation of AMPs as capture elements in a biosensor stemmed from U.S. Army research laboratories [58,59]. AMPs were immobilized either in wells for colorimetric revelation with horseradish peroxidase [58], or on regular microscope slides for measurements with a fluorescence sensor [59]. The fluorescence-based biosensor of Kulagina and colleagues was subsequently extended to other targets, such as viruses and bacteria, by incorporating novel peptides [60,61]. The lowest detection limits then obtained were about 10^4^ CFU·mL^−1^ for the detection of *E. coli* with magainin [60].

The sensor that was developed, as shown in Figure 2, required the use of antibodies or fluorophore-labelled bacterial cells to reveal the presence of bacteria. Such technique makes it non-suitable for wide spectrum applications due to the intrinsic highly specific nature of antibodies. Arcidiacono and colleagues have provided the first example of direct detection of bacteria with peptides labelled with a fluorescent marker (cyanine 5) [29]. They obtained a high specificity for *Escherichia coli* O157:H7.

### 3.2. Improving Biosensors Operability by Designing Label-Free Approaches

The first study demonstrating the label-free detection of microbial cells oby a biosensor thanks to AMPs was achieved in 2009 by Zampa and colleagues [62]. The *Leishmania chagasi* parasite was detected with a limit of detection of 10^3^ cells·mL^−1^, thanks to dermaseptin-01 AMPs immobilized in electroactive nanostructured layered films. Measurements were made by cyclic voltammetry, thus proving the usability of unlabelled AMPs in biosensors. Such approach brings several benefits as it usually simplifies the detection process. Indeed, no washing steps are required and this significantly reduces the cost, since labels, such as fluorophores or magnetic particles, are quite expensive reagents. In the case of peptide-based label-free detection of bacteria, Mannoor et al. issued the first system in which an AMP (namely maganin I) was used as biorecognition element [31]. This proof of concept of the specific recognition of Gram-negative pathogens by magainin I was carried out on *Escherichia coli* O157:H7 and *Salmonella typhimurium* strains. The technique consisted in immobilizing the AMP onto an array of interdigitated microelectrodes suitable for an electrical readout (see Figure 3). The detection limit reached is then of the order of 10^3^ CFU·mL^−1^ in a saline solution. This seminal work opened the breach to numerous electrochemical biosensors, and more generally, AMP-based biosensors for the detection of bacteria. The next part will give an overview of the state of the art of such sensors, including the publications issued in the field after the pioneering work of Mannoor et al. published in 2010.

### 3.3. Biosensors Based Solely on AMPs for the Recognition of Bacteria

The first applications of AMPs for the detection of bacteria are fairly recent (about 15 years old), nonetheless they have been adapted to a large variety of contexts and devices. This versatility comes from their ease of chemical modification which makes AMPs suitable for integration within pre-existing technologies. In the following section, we propose to review the state of the art of label-free biosensors for the detection of bacteria using AMPs as unique recognition elements. Uses where peptides play an auxiliary role to detection, such as capture prior to label mediated-detection of pathogens by spectroscopy [63,64,65] or with PCR [66] were ruled out. Only biosensors where the analysis chain is fully integrated are considered, which excludes applications based on the combined use of AMPs and labels with subsequent analysis techniques such as fluorescence or microscopy [48,66,67,68,69,70,71]. In other words, this means that the result must be interpretable by the operator without any additional handling step other than that of bringing the sample into contact with the biosensor. Although in some cases sample preparation may be required, for instance through concentration with magnetic beads [72], the recognition element remains the AMP. In addition, only applications targeting whole bacteria are considered, which is the closest to real-world considerations, where only viable bacterial cells are looked for. It should be noted, however, that many of these publications use bacteria that are inactivated whether chemically, by radiation or heat. A summary of the main characteristics of such biosensors from the state of the art is available in Table 2. Information such as durations of the assay, volumes or flow rates are given, along with the names of the AMPs that are used and the techniques employed.

After their first proof of concept, Mannoor and colleagues have succeeded themselves by designing the first AMP-based flexible electronic sensor, which was implantable on teeth or inside intravenous bags [34]. Contamination of the latter can be followed in a wireless fashion, easing the monitoring of the health status of patients. Based on resistivity changes, the sensor is made out of reduced graphene on which the odorranin-HP AMP is grafted thanks to its coupling with a graphene bonding peptide. Performances of the device allowed the detection of *E. coli* and *S. aureus* at a concentration of 10^3^ CFU·mL^−1^ in PBS. In a more complex medium, such as saliva, *Helicobacter pylori* presented a limit of detection of 10^4^ CFU·mL^−1^. Such results are promising for an easy integration of AMP-based biosensors in various set-ups, even directly on tissues suspected to be infected, allowing to perform a ubiquitous wide-spectrum detection of pathogens.

These first proofs of concept are the precursors of numerous subsequent articles using electronic or electrochemical transduction devices with antimicrobial peptides. Hoyos-Nogués et al. discussed these electrochemical biosensors using AMPs with more details in a recent review [73]. Nonetheless, antimicrobial peptides have been used with several other transduction methods. Besides the works by Mannoor and colleagues, another example of integration on a graphene transistor has been described [74]. Chen et al. used the same AMP as Mannoor et al. (magainin I), and the detection threshold was in that case below 10^3^ CFU·mL^−1^. Such achievement emphasizes that biosensors can be improved not only by bettering the immobilization of the probe, but also by tailoring a device with higher sensitivity.

Shi and colleagues have been, to the extent of our knowledge, the first ones to confirm the ability of AMPs to maintain their recognition performances in blood [75]. The principle of their piezoelectrical sensor relies on the release of adsorbed pleurocidin AMPs in solution when bacteria are present. Limits of detection down to 10 CFU·mL^−1^ are reached in 50% sheep blood for *E. coli*, in less than 15 min. Other pathogens such as *Klebsiella pneumoniae*, *Enterococcus faecalis* or *Staphylococcus aureus* are detected down to 10^2^ CFU·mL^−1^. This concentration is close to the ones found in medical samples during bacterial infections. However it remains one order of magnitude higher than the usual bacterial concentration in blood during infections, which is considered to be around 1 CFU·mL^−1^ [76]. Besides this, the release of the AMP may potentially kill the planktonic bacteria before the limit of detection is reached, thus giving a false negative result. Furthermore, no identification of the pathogen is given by the technique, although it could be adapted in a multiplexed fashion with a combination of several sensors bearing AMPs that have various spectra of affinity towards bacteria. Nonetheless, its limit of detection is low enough to consider using it after or during a short enrichment step, thus giving faster information about the presence or absence of pathogens in a sample. This study was also a proof of the operability of AMPs as probes in complex media, which was also confirmed in other samples, such as diluted milk [30] or ground beef [71].

Biosensors such as the ones proposed by Mannoor et al. [31] or Etayash et al. [30,77] are able to distinguish large groups of bacteria, based on their Gram staining for instance. In the first case, this discrimination ability has been explained by the propensity of the magainin I AMP to bind preferentially to the outer membrane of Gram-negative bacteria, due to the directly accessible phospholipids in the outer membrane [31]. Indeed, the presence of LPS and thus long O-antigens favours the electrostatic interactions and hydrogen bonding [78]. Regarding the studies led by Etayash et al., the incorporated peptide was either leucocin A or fragments thereof [30,77]. Such AMPs from the bacteriocin class are known to be targeting the mannose-phosphotransferase system (man-PTS) as a docking site on the bacterial outer membrane [79]. The prevalence of man-PTS in Gram-positive bacteria could hence explain the specificity of the sensor developed by Etayash et al. These examples underline the absence of any general rule regarding the specificity of a peptide incorporated in a sensor. Each AMP may behave differently, thus needing a characterization of its activity-spectrum. This emphasizes the need to properly choose the AMP probe and sensing devices depending on the targeted application–specific detection in a complex matrix will not require the same specifications as universal sensing in simple culture medium.

Other biosensors, based on electrochemical methods, such as those from Andrade et al. [80], Miranda et al. [81] or Junior et al. [82] provide identification of the Gram staining of unknown pathogens, using nanostructured electrodes and electrochemical impedance spectroscopy (EIS). These three studies were based on the use of clavanin A, an AMP known for its wide-spectrum of activity directed against both Gram-positive and negative species [83]. Nonetheless, higher affinities towards Gram-negative bacteria were observed. This difference is hypothesized to come from the dissimilarities between the cell walls of Gram-positive and negative bacteria [80,82]. The latter having an outer membrane that is more negatively charged, due to the abundance of anionic LPS. Preference for Gram-negative species would hence be explained by the predominance of electrostatic and hydrophobic interactions [17,84], however this identification of the Gram qualification implies the need to know the bacterial concentration in the sample, which is unlikely in real-world applications for the detection of infections. The work of Wilson et al. [72] is also based on EIS and gave similar results, still with an identification capacity which is highly dependent on the knowledge of the bacterial concentration as a prerequisite. Furthermore, these EIS devices are usually highly sensitive to the composition of the medium. For instance, the limit of detection reported by Miranda et al., is 10 CFU·mL^−1^ in a 2 µL volume, thus meaning they are able to detect the presence of virtually 0.02 CFU in the sample. Such sensitivity raises wonders about the system’s compatibility in highly varying biomedical samples as blood, saliva or urine can be.

In a recent study [85], we proposed an approach inspired from a strategy proven to be able to detect pathogens at low concentrations, even in complex matrices [86,87,88]. This assay, based on surface plasmon resonance (SPR) imaging enables the monitoring of growing pathogens, thanks to the “Culture-Capture-Measure” method. Virtually, any viable bacterium is detectable in this setting, the limiting factor being the pathogen ability to grow and the delay before the enrichment surpasses the limit of detection. The multiplexing ability allows to use several probes in parallel: 6 AMPs were thus assayed against 5 different common pathogens, namely *Escherichia coli*, *Staphylococcus aureus*, *Salmonella, Staphylococcus epidermidis* and *Listeria,* with respective initial concentration of 51, 16, 6, 2.5 × 10^3^ and 2.6 × 10^3^ CFU·mL^−1^. Multivariate analysis methods have made possible the classification specie by specie of the obtained response profiles. The combination of wide spectrum AMPs that have various ranges of affinity makes it possible to consider the response of the sensor as a whole, rather than examining individual outputs of each AMP. This work constitutes a base for further work in order both to improve the affinities of the AMPs towards bacteria but also to better the sensing system with a higher sensitivity and upgraded subsequent statistical analyses. Future works will also include detection in complex matrices such as blood.

## 4. Perspectives and Outlook

The state of the art depicted in Table 2 represents the fast-paced evolution of AMP-based label-free biosensors. Taking advantage of the progresses made in the overall field of detection devices, an interesting development pathway seems to be ahead. Compared to biosensors incorporating other kinds of probes, AMP-based biosensors are now demonstrating similar performances. For instance, the electrochemical assay designed by Wilson et al. [72] has similar features compared to other electrochemical biosensors operating in food [96]. Alternatively, some systems based on AMPs such as the SPR imaging device we designed [85] do not present faster detection times than previous studies [86,87,88]. However, it is able to detect a broader range of bacteria than devices based on antibodies. This highlights the need to design assays according to the requirements of the targeted practical implementations. Despite this progress, some of the requirements to achieve real-world applications are indeed still lacking. Advances such as the design of improved sensors, the incorporation of nanomaterials or the exploration of the chemical space offered by AMPs are some of the evolutions we can expect.

### 4.1. Advantages and Limitations of AMP-Based Biosensors for the Detection of Pathogens

Detection of pathogenic bacteria is still largely based on cultural methods and mass spectrometry, despite the rise of molecular methods [97]. In this context, biosensors based on AMPs are not only challenging the biosensors based on other types of probes such antibodies or aptamers but also the pre-existing techniques. When one considers a given application, evaluation of AMP-based systems should thus be made in comparison against other methodologies. Indeed, real-world situations in which pathogens are to be detected offer various harsher conditions compared to ideal conditions in state-of-the-art research laboratories. Bacterial concentrations to be detected are in the range of a few colony forming units per sample [76]. The media in which the assays are led are often complex matrices containing interfering molecules, such as blood or food. As underlined in the review by Rajapaksha et al., there are several requirements a detection method should include [98]. Specificity, reliability, rapidity and potential for high throughput analysis are some the main aspects. However, characteristics such as being low cost, easily available and standardized are also of first importance. Moreover, the assessment of the viability of the pathogens is sometimes needed too [99]. More complex contexts, such as bloodstream infections, require additional features including the identification of polymicrobial infections or the detection of drug resistance [25]. To the extent of our knowledge, there is hitherto no single system possessing all these features. We propose in Table 3 a non-exhaustive comparison of the main techniques used in pathogen detection in clinical settings–during bloodstream infections for instance. For most of the entries, a review by Guido et al. gives an overview of the commercially available systems [100]. State-of-the-art reviews can also give an in-depth understanding of the methods used to detect bacteria in a clinical context [24,101,102,103].

From the overview given in Table 3, it appears that no method is currently satisfying all needs required for the detection of bacteria. Hence, a combination of the mentioned techniques is used in clinical laboratories. In this context, microarrayed label-free biosensors–potentially based on AMPs–could perform efficiently in pairing with conventional methods. They present a high potential for miniaturization, as well as a promising integrability. Various means of improvement are possible. On the first hand, advances in tunability through a better understanding of the recognition mechanisms of AMPs could allow smarter designs of biosensors. On the other hand, progresses in the development of new devices are of great interest to improve their sensitivity and integrability.

### 4.2. Exploiting the Versatility of AMPs

Label-free biosensors based on AMPs as only recognition elements have so far used only an insignificant number of known AMP sequences. About twenty peptides are included in the studies reported in Table 2, compared to several thousands of entries in online databases [104,105]. Such gap highlights the unexploited potential of these molecules. The progress of biosensors towards high-throughput devices with enhanced multiplexing abilities is of great interest to screen more AMPs as probes [106]. More sequences could thus be assayed. Furthermore, peptide engineering could also improve the affinity of surface-tethered peptides towards bacteria. Techniques such as phage-display could also help design peptides with a high specificity against certain pathogens [107]. In silico studies also hold the prospect of new advances in the design of bio-active peptides [108]. Overall, progresses in the understanding of structure-activity relationship of AMPs would be a payoff both for drug development and biosensing devices [9,20,44].

The primary structure of a peptide is however not the only factor influencing its activity. The chemical versatility of peptides allow to tune their orientation onto surfaces [53,55,109]. The same sequence can thus yield very different results depending on its immobilization strategy [55]. Beyond the orientation, variations of the deposition method also produce different performances for peptides on a sensor surface [110]. The transposition of such optimization techniques from other fields of applications to label-free bacterial biosensors would be a way towards the improvement of their performances.

AMP-based biosensors can therefore be improved whether by the design of new probes or thanks to enhancements of the methodologies employed for surface functionalization and characterization. This potential, coupled to the aforementioned versatility of AMPs, is still promising a wide range of applications, either as universal biosensors or as specific devices targeting peculiar pathogenic agents.

### 4.3. Emerging Trends in the Design of Biosensors

Performances of biosensors are not only based on the characteristics of the incorporated recognition elements. Several other elements can be improved to increase the sensitivity and efficiency of biosensing devices.

First and foremost, sensitivity of biosensors can be improved through optimizations of the transduction mechanism. Technical progresses in recent years have given researchers several tools to achieve this goal. The rise of better electronic devices, along with the emergence of nanomaterials and nanofabrication, has overall enhanced the characteristics of biosensors [96,111,112]. Higher resolutions and sensitivities can now be achieved. The inclusion of nanomaterials such as nanostructured surfaces or nanoparticles has generated a great deal of interest in the biosensing community [113,114,115]. The addition of new physical and chemical properties, as well as the increase of the specific surface area, opens up now possibilities. Amplification of the signal can thus be obtained by decorating nanoparticles with ligands. AMP-based biosensor using this principle have already been studied by de Miranda et al. [81] and Silva Junior et al. [82]. Inclusion of gold nanoparticles (AuNPs) yielded a 10-fold improvement of the limit of detection compared to previous studies employing the same system without AuNPs [80]. Contributions of such nanomaterials can therefore provide substantial improvements to the performances of AMP-based biosensors.

Furthermore, when it comes to the detection of small bacterial concentrations, for instance in foodstuff or during bloodstream infections, another improvement can be the integration of enrichment steps. Generally based on the capture and subsequent concentration of bacteria, several techniques exist. Magnetic beads or nanoparticles decorated with ligands targeting bacteria are a first solution [116,117]. Passive solutions such as nano-patterned trapping surfaces or physical filters can also help extract bacteria from complex samples [118,119,120]. Capture surfaces decorated with ligands are another way which is close to the one adopted in biosensors [121]. Acoustophoresis using sound waves to isolate bacterial cells or probes inside samples has also been studied lately [90,122]. It is of note that these techniques are not self-exclusive and can therefore be combined inside the same device. A review by Sande et al. gives an overview of the recent advances concerning the extraction of bacteria from complex samples [123]. Burklund and Zhang issued a review covering specifically the case of pathogen extraction from blood [124]. Integration of such techniques has also been eased thanks to the evolution of microfluidics. Several steps of an assay can now be performed in a single-chip, thus allowing the chaining of pre-enrichment and analysis. Recent reviews give a good overview of the developments of these techniques [125,126]. However, the size of the analysable samples in microfluidic chips can be limiting. Recovery of bacteria from the bloodstream for instance requires volumes ranging up to 60 mL [76]. Nonetheless, cultural enrichment of samples prior to pre-concentration could allow shorter assay times.

Development of commercial label-free biosensors based on AMPs for the detection of pathogenic bacteria will definitely require one to harness the complementarity of all available techniques. Integration of such sensors in pre-existing routines in microbiological laboratories could also be another path to ensure wide adoption.

## 5. Conclusions

In a context of growing global bacterial resistance to drugs, better and more rapid detection of pathogenic threats is one of the biggest challenges we are facing. The recent advances carried in the field of biosensors contribute to the efforts made to face this issue. The development of better diagnostic systems based on biosensors implies more efficient sensing devices and also adequate biorecognition elements, which would ideally allow the specific identification of any pathogen, particularly in medical settings where the nature of the threat is often ambiguous.

In this review we mainly focused on summarizing the state of the art of antimicrobial peptides as the main biorecognition element in label-free biosensors. The integration of AMPs in continuously evolving biosensing technologies opens up the perspective of faster and more efficient diagnostic systems. On the first hand, thanks to their wide spectra of interactions with bacteria, AMPs are enabling either specific identification or universal detection, depending on the nature of the chosen AMPs and the mode of tethering. On the other hand, the ease of synthesis and chemical modification of peptides, along with their high stability make them ideal candidates to be incorporated in various devices, thus showing great versatility compared to conventional proteinic probes. Numerous parameters can be tuned to achieve improved performances for AMP-based sensors: the design of the AMP itself, the sensitivity of the biosensing system or the methodology by which the sensor is exposed to the sample.

Notwithstanding their potential, there are no commercial diagnostic devices based on antimicrobial peptides yet. Moreover, the number of different AMPs that were assayed hitherto (a few dozen only) is insignificant compared to the thousands of discovered sequences. In spite of these observations, we have shown both the adaptability and potential of AMP-based biosensors. Future research in the field will probably unveil promising results, leading to the application of such biosensors in real-world contexts, for instance in clinical laboratories as well as in the drug or food industries.

## Figures and Tables

**Figure 1 molecules-25-01998-f001:**
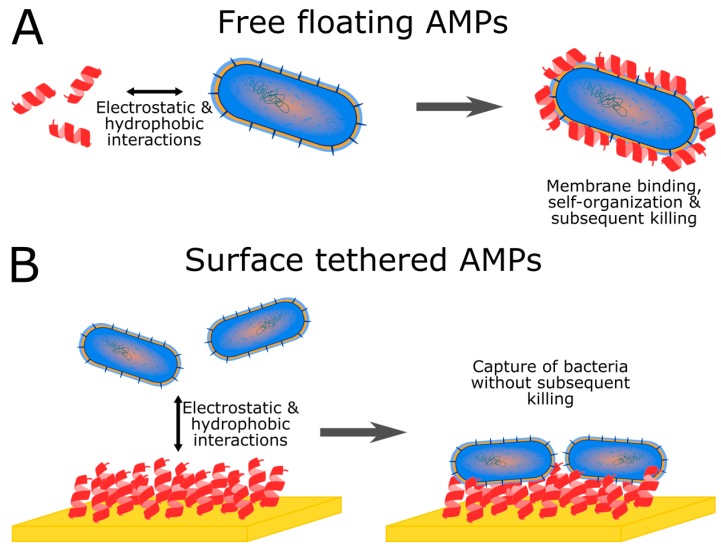
Tethering AMPs can prevent the triggering of killing mechanisms of the bacteria. (**A**) Free floating peptides can self-organize onto the bacterial membrane and subsequently disrupt it or insert themselves inside the cell in order to kill it. (**B**) Tethered peptides can no longer self-organize at the surface of the bacterial membrane, thus inhibiting their bactericidal activity. Interaction abilities can, however, still be conserved, thus allowing the design of capture surfaces for biosensing purposes.

**Figure 2 molecules-25-01998-f002:**
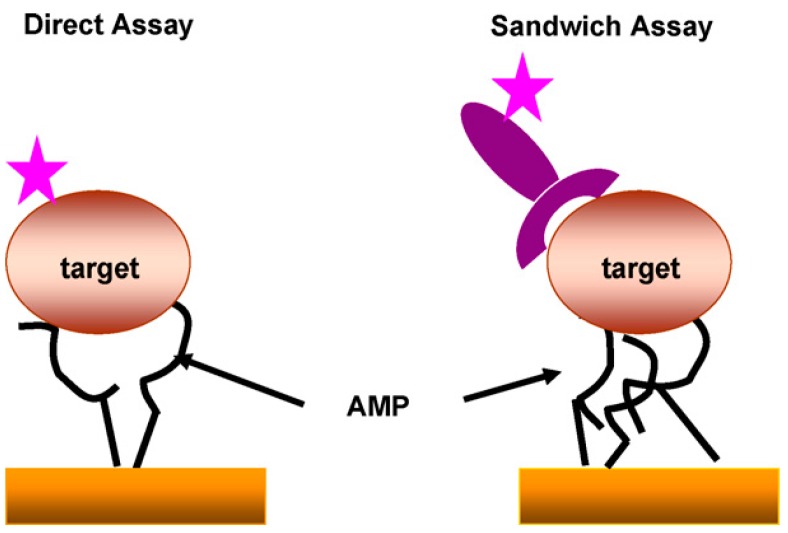
Schematic principle of the assays developed by Kulagina et al. in which the antimicrobial peptides were covalently immobilized on a surface, thus acting as capture molecules binding to bacteria. The detection itself is performed using fluorescent labelling, either of the bacteria directly or through a specific fluorescing antibody. Reproduced from [60].

**Figure 3 molecules-25-01998-f003:**
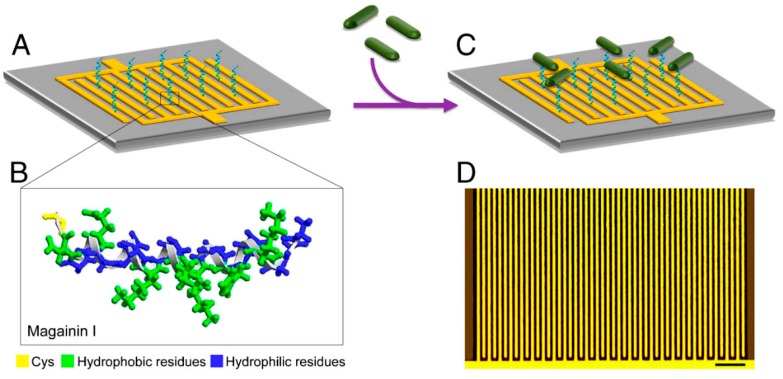
AMP-based device for the electrical detection of bacterial pathogens. (**A**) Schematic of the interdigitated microelectrode array, with immobilized AMPs. (**B**) Representation of the magainin I in helical form, with the added terminal cysteine residue, allowing its anchoring on gold. Hydrophobic and hydrophilic residues are highlighted to emphasize the amphiphilic nature of magainin I. (**C**) Binding of bacterial cells on the array, thus enabling the detection. (**D**) Optical image of the interdigitated microelectrode array (scale bar: 50 µm). Reproduced from [31].

**Table 1 molecules-25-01998-t001:** Examples of antimicrobial peptides used in biosensors along with their sequences and reported specificities.

Peptide	Sequence	Reported Specificity ^1^	Ref.
C16G2cys	TFFRLFNRSFTQALGKGGGKNLRIIRKGIHIIKKYGGGC	*Streptococcus mutans*	[28]
Cecropin P1	SWLSKTAKKLENSAKKRISEGIAIAIQGGPR	*Escherichia coli* O157:H7	[29]
G10KHc	KKHRKHRKHRKHGGSGGSKNLRRIIRKGIHIIKKYGC	*Pseudomonas aeruginosa*	[28]
Leucocin A	KYYGNGVHCTKSGCSVNWGEAFSAGVHRLANGGNGFW	Gram-positive species	[30]
Magainin I	GIGKFLHSAGKFGKAFVGEIMKS	Gram-negative species	[31]
		*E. coli* O157:H7	[32]
MSal 020417	NRPDSAQFWLHHGGGSC	*Salmonella* spp.	[33]
Odorranin-HP	GLLRASSVWGRKYYVDLAGCAKA	Broad-spectrum activity	[34]
Synthetic peptide	WK_3_(QL)_6_K_2_G_3_C	Broad-spectrum activity	[35]

^1^ Indicates the sensor specificity claimed in each study.

**Table 2 molecules-25-01998-t002:** Biosensors based only on AMPs for recognizing bacterial targets. Articles are classified chronologically.

Peptide	Target	Threshold (CFU·mL^−1^)	Duration	Volume/Flowrate	Medium	Transduction Mechanism	Ref.
Magainin I	*E. coli* O157:H7	10^3^	20 min	5 µL·min^−1^	PBS	EIS	[31]
	*S. typhimurium*	10^4^					
Odorranin-HP	*E. coli; S. aureus*	10^3^	30 min	1 µL	PBS	Resistive sensor made in graphene. Biocompatible and wireless communication	[34]
	*H. pylori*	10^5^	10 min	1 µL	Saliva		
Magainin I	*E. coli* O157:H7	10^3^	90 min	-	PBS	EIS	[89]
Leucocin A	*L. monocytogenes; S. aureus; E. faecalis; L. innocua*	10^3^	20 min	20 µL	PBS Milk:PBS (1:9) (only for *Listeria*)	EIS	[30]
G10KHc C16G2cys	*P. aeruginosa; S. mutans*	10^5^	25 min	A few microlitres	Saline buffer	Microfluidic chip coupled to EIS	[28]
MSal 020417 (phage-derived peptide)	*Salmonella* spp.	10^6^	< 10 min	25.2 µL·min^−1^	PBS	Micro-cantilevers	[33]
Indolicidin	*E. coli* O416	10^5^	2 min	15 µL·min^−1^	PBS	Fluorescently labelled AMPs monitored thanks to UV in a microfluidic chip	[90]
		10^8^			Tap water		
Magainin I	*E. coli* O157:H7	10^4^	60 min	50 µL·min^−1^	PBS	Conductimetry measurement on completely reduced graphene oxide transistors	[74]
Clavanin A	*E. faecalis; E. coli; B. subtilis; K. Pneumoniae*	10^2^	10 min	1 µL	PBS	EIS sensor using carbon nanotubes structuration	[80]
Magainin I	*E. coli* O157:H7	4 × 10^2^	10 min	-	PBS	QCM	[32]
		1.5 × 10^3^	10 min	-	PBS	EIS	
Leucocin A Leu10 (a leucocin A fragment) Ped3 (a pediocin fragment)	*L. monocytogenes*	10^5^	60 min	5 mL·h^−1^	PBS	Micro-cantilevers	[39]
Colicin V	*E. coli* O6	10^2^	A few minutes	A few microlitres	PBS	EIS	[91]
Magainin I	*E. coli* O157:H7	1.2 × 10^2^	30 min	100 µL	PBS	Electrochemiluminescence amplified by a ruthenium-magainin I complex	[92]
WK_3_(QL)_6_K_2_G_3_C	*E. coli; S. aureus; P. aeruginosa; S. epidermidis*	10^2^	30 min	100 µL	Tris-HCl	EIS	[35]
Pleurocidin	*E. coli*	10	< 15 min	2 mL	Sheep blood 50% SPS 0.01 %	Piezoelectrical sensor	[75]
	*E. faecalis; S. aureus; P. aeruginosa; K. pneumoniae; E. cloacae; C. albicans*	10^2^					
Human Lactoferrin (residues 1 to 11)	*S. sanguinis*	3.5 × 10^1^	30 min	100 mL	*KCl*	EIS	[93]
		8.6 × 10^2^			Artificial saliva		
Clavanin A	*E. coli; S. typhimurium; E. faecalis; S. aureus*	10	70 min	2 µL	PBS	EIS	[81]
Magainin I	*E. coli* O157:H7	5 × 10^2^	10 min	200 µL	Water; apple juice; orange juice; mixed fruit and vegetable juices	SPR on fibre bundles amplified with silver nanoparticle-reduced graphene oxide nanocomposites	[94]
Clavanin A	*E. coli; S. typhimurium; E. faecalis; S. aureus; K. pneumoniae; B. subtilis*	10	-	2 µL	PBS	EIS	[82]
Paired fragments of Leucocin A	*L; monocytogenes*	10	60 min	2 mL	Sea water	One fragment is coupled to magnetic beads for isolation. The other is coupled to HRP for potentiometric spectroscopy.	[95]
Melittin	*E. coli* O146	1 (or 3.5 for apple juice)	25 min	250 µL (20 µL are needed for a measurement)	PBS; drinkable water & apple juice (only for *E. coli*)	EIS and peptide covered magnetic beads for concentrating bacteria	[72]
	*S. typhimurium; S. aureus*	10					
Clavanin A; Magainin I; Ped3; PGQ; Leucocin A24	*S. typhimurium*	6	9 h	1 mL	TSB	SPR imaging of living bacteria cultures	[71]
	*S. aureus*	16	7 h				
	*E. coli* O1:K1:H7	51	11 h				
	*S. epidermidis*	2.5 × 10^3^	6 h				
	*L. monocytogenes*	2.6 × 10^3^	19 h				

The reported threshold values are the lowest concentrations that were detected by the biosensors in each study. EIS: Electrochemical Impedance Spectroscopy; HRP: Horse-Radish Peroxidase; PBS: Phosphate Buffered Saline; TSB: Tryptic Soy Broth; SPR: Surface Plasmon Resonance.

**Table 3 molecules-25-01998-t003:** Methods for the detection of pathogenic bacteria in clinical settings along with respective main advantages and drawbacks.

Method	Advantages	Drawbacks
Conventional culturing methods	High reliability when thoroughly performed Simplicity of protocols Can indicated the contamination level in samples Some instruments are now largely automated (but therefore are no longer low-cost)	Time consuming (up to 7 days) Requires one to work in aseptic conditions: high risk of environmental contamination Needs trained staff Impossible to detect emerging and non-culturable pathogens
Polymerase Chain Reaction (PCR)	Rapid turnaround Can be multiplexed to target different genes Reliable in cases of high levels of contamination	Most tests do not distinguish live and dead bacteria Protocols need a high level of expertise for the handling Possible presence of PCR inhibitors in some matrices Only pathogens with known sequence data can be detected Contamination can lead to confusing results Instruments and consumables can be expensive
Mass Spectrometry	Rapid turnaround High throughput Low cost for single analyses (but expensive device)	Difficult to directly use raw samples (enrichment or extraction of bacteria is often needed) Only pathogens with known fingerprints can be reliably identified
Optical biosensors	Able to detect low bacterial concentrations in a rapid fashion Information is both quantitative and qualitative Protocols are simple and samples do not require laborious preparative steps	A lot of handling is required thus needing trained staff Low throughput
Label-free biosensors	High sensitivity Tunability of the specificity by tailoring ligands Easily automated and interpreted Some systems can assess the viability of bacteria Easily miniaturized: integrable in pre-existing routines and devices	Requires to develop adequate ligands Throughput depends on the system Scalability towards commercial systems still not assessed Varying cost depending on the technology

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
