# Peer review of "Antimicrobial Peptides as Probes in Biosensors Detecting Whole Bacteria: A Review"

_molecules, 2020, doi:10.3390/molecules25081998_

Round 1

Reviewer 1 Report

The review presented by Pardoux is well written although it is focused on a too young approach with a limited number of references. There are only few mistakes or sentences too many long. Overall this manuscript, to be considered as a mini-review, is acceptable after a minor revision of the errors in the text.

Few considerations:

Table 1, row 9: regarding magainin 1, the threshold value associated to E.coli O157:H7 is ambiguous.... Is it correct? Same situation in the last row: 6-16-51 values. Could these values be indicated otherwise?

Page 3, line 126. Maybe « this » is more suitable than « that » at the beginning of the sentence

Paragraph 2.3 page 4. The paragraph ends without an appropriate reference

Page 6 line 240 It would be advisable to insert the reference at the end of the sentence

Reviewer 2 Report

Recommendations in attached file

Round 2

Reviewer 2 Report

Can be accepted for further publication steps.